# Elevated glutamate impedes anti-HIV-1 CD8 + T cell responses in HIV-1-infected individuals on antiretroviral therapy

You-Yuan Wang[1,2,6], Cheng Zhen[2,6], Wei Hu[3,6], Hui-Huang Huang[2], Yan-Jun Li[4], Ming-Ju Zhou[2], Jing Li[2], Yu-Long Fu[2], Peng Zhang[2], Xiao-Yu Li[2], Tao Yang[1,2], Jin-Wen Song [1,2], Xing Fan[2], Jun Zou[4], Si-Run Meng[4], Ya-Qin Qin[4], Yan-Mei Jiao[2], Ruonan Xu[1,2], Ji-Yuan Zhang[2], Chun-Bao Zhou[2], Jin-Hong Yuan[2], Lei Huang[2], Ming Shi[1,2], Liang Cheng[5], Fu-Sheng Wang [1,2,4✉] & Chao Zhang [1,2,4✉]

CD8 + T cells are essential for long-lasting HIV-1 control and have been harnessed to develop therapeutic and preventive approaches for people living with HIV-1 (PLWH). HIV-1 infection induces marked metabolic alterations. However, it is unclear whether these changes affect the anti-HIV function of CD8 + T cells. Here, we show that PLWH exhibit higher levels of plasma glutamate than healthy controls. In PLWH, glutamate levels positively correlate with HIV-1 reservoir and negatively correlate with the anti-HIV function of CD8 + T cells. Single-cell metabolic modeling reveals glutamate metabolism is surprisingly robust in virtual memory CD8 + T cells (TVM). We further confirmed that glutamate inhibits TVM cells function via the mTORC1 pathway in vitro. Our findings reveal an association between metabolic plasticity and CD8 + T cell-mediated HIV control, suggesting that glutamate metabolism can be exploited as a therapeutic target for the reversion of anti-HIV CD8 + T cell function in PLWH.

[1] Medical School of Chinese PLA, Beijing, China. [2] Department of Infectious Diseases, The Fifth Medical Center of Chinese PLA General Hospital, National Clinical Research Center for Infectious Diseases, Beijing, China. [3] Department of Emergency, Fifth Medical Center of Chinese PLA Hospital, Beijing, China. [4] Guangxi AIDS Clinical Treatment Centre, The Fourth People's Hospital of Nanning, Nanning, China. [5] Medical Research Institute, Frontier Science Center of Immunology and Metabolism, Zhongnan Hospital of Wuhan University, Wuhan University, Wuhan, China. [6]These authors contributed equally: You-Yuan Wang, Cheng Zhen, Wei Hu. ✉email: fswang302@163.com; zhangch302@163.com

Despite considerable progress in antiretroviral therapy (ART) for people living with HIV-1 (PLWH), HIV-1 cure remains elusive. ART cannot eliminate the viral reservoir, driving the resurgence of infection on discontinuation. Immunological approaches, such as checkpoint inhibitors, therapeutic vaccines, and neutralizing antibodies, hold promise for HIV-1 cure[1]. However, knowledge gaps remain regarding how HIV-1 infection shapes the host immune system and immune responses that effectively control the virus.

The critical role of CD8+ T cells in controlling primary HIV-1 infection has been well established. However, until recently, CD8+ T cells were recognized as essential for long-lasting HIV-1 control in ART-treated individuals. Strong evidence comes from the observation that the deletion of CD8 + T cells leads to a viral rebound in SIV-infected ART-treated macaques[2,3]. CD8+ T cells exert anti-HIV-1 functions via both cytolytic (granzyme and perforin) and non-cytolytic (cytokines and beta-chemokines) mechanisms[4]. During chronic HIV-1 infection, the anti-HIV-1 functions of CD8+ T cells are impaired, termed functional exhaustion[5], and are difficult to fully restore even after long-term ART[6]. However, early ART initiation[7,8] or combined treatment with broadly neutralizing anti-HIV-1 antibodies at ART initiation[9], effectively preserves the function of CD8+ T cells, coinciding with the small HIV-1 reservoir size in PLWH. To date, there have been no successful attempts to harness CD8+ T cells for functional cure in PLWH. For example, treatment with latency-reversing agents enhanced HIV-1 transcription but failed to shrink the viral reservoir size[10,11]. Moreover, T-cell therapeutic vaccines successfully induced HIV-1-specific CD8+ T-cell responses but showed limited impact on viral reservoir size and viral rebound in PLWH[12,13]. Thus, a better understanding of the antiviral activities of CD8+ T cells is needed for immune interventions aimed at harnessing these functions to target the HIV-1 reservoir.

Metabolic profiling is an important determinant of CD8+ T cell anti-HIV-1 function[14]. Long-term HIV-1 infection and prolonged ART use often cause metabolic-related diseases[15] and significantly impact the immune system[16,17]. An in vitro study compared the metabolic profile of CD8 + T cells in PLWH on ART with that of CD8+ T cells from elite controllers (ECs) and found that the former is characterized by less metabolic plasticity and antiviral function but more susceptible to metabolic intervention[18]. Furthermore, the altered metabolic status of PLWH and the dynamics of the HIV-1 reservoir can be visualized in the changes in metabolite levels[19,20].

We recently identified virtual memory T (TVM) cells, a particular subset of CD8 + T cells characterized by antigen-naïve but possessing a memory phenotype[21,22], as an immunological determinant factor for HIV-1 reservoir size[23]. TVM cells can sense and inhibit HIV-1-replicating cells via HLA/KIR signaling[23]. Among the multiple cytokines highly expressed by TVM cells, we identified CCL5 as a critical effector molecule in the control of the HIV-1 reservoir[24]. Furthermore, TVM cells display a unique mixed effector-memory metabolic characteristic that can relies on glycolysis and oxidative phosphorylation[25]. Whether metabolism-related factors regulate the anti-HIV-1 function of TVM cells in PLWH is unclear.

In this study, we aimed to understand the changes in metabolites and their relationship with the anti-HIV-1 function of CD8+ T cells in PLWH. First, we analyzed the associations among plasma metabolites, HIV-1 reservoir size, and the anti-HIV-1 effector function of CD8+ T cells in PLWH on ART. Furthermore, we profiled the metabolic characteristics of CD8+ T cells from PLWH at the single-cell transcriptional level. Finally, we used an ex vivo assay to confirm the effects of glutamate, the most relevant metabolite for the HIV-1 reservoir, on CD8+ T cell and TVM cell function and clarified the crucial mechanisms involving the mechanistic target of rapamycin complex 1 (mTORC1) pathway. This finding reveals that elevated plasma glutamate inhibits the anti-HIV-1 function of CD8+ T cells and is associated with a larger HIV-1 reservoir size in PLWH.

## Results

**Glutamate levels were positively associated with the size of the HIV-1 reservoir in PLWH.** HIV-1 infection induces remarkable alterations in the metabolic state of the body[26,27], but nevertheless, the specific effect of the metabolic changes on the HIV-1 reservoir and anti-HIV function of CD8+ T cells remains unclear. To address this question, we used a liquid chromatography-mass spectrometer (LC-MS) to quantify multiple potential metabolites in plasma from a previously established ART cohort (Fig. 1a). We further investigated the association between HIV-1 reservoir size and the functionality of CD8+ T cells with metabolite levels using correlation analysis (Fig. 1a). Furthermore, ex vivo experiments were conducted to clarify the mechanism by which the metabolites affect CD8+ T cell function (Fig. 1a).

Overall, we identified 12 metabolites in four groups: five short-chain fatty acids (SCFAs), two tryptophan metabolites, two glycans, and three energetic metabolic substrates. Among all metabolites detected, the levels of acetate, α-ketoglutaric (α-KG), pyruvic acid, and glutamate were significantly higher in PLWH ($n = 59$) than in healthy controls (HCs, $n = 11$) (Supplementary Table 1). In contrast, oxindole, fucose, and GlcNAc were less abundant in PLWH than in HCs (Fig. 1b). Correlations between HIV-1 reservoir size and metabolites that exhibited alterations in PLWH were analyzed. The levels of GlcNAc and fucose negatively correlated with HIV-1 cell-associated unspliced RNA (CA usRNA) (GlcNAc, $r = -0.345$, $P = 0.007$; fucose, $r = -0.326$, $P = 0.012$) and tended to be negatively correlated with HIV-1 DNA (Fig. 1c, d). Acetate in the SCFA group positively correlated with HIV-1 CA usRNA ($r = 0.325$, $P = 0.0012$) and tended to be positively correlated with HIV-1 DNA. Three energy metabolism substrates showed significant positive correlations with HIV-1 CA usRNA: α-KG ($r = 0.4693$, $P < 0.001$), pyruvic acid ($r = 0.5830$, $P < 0.001$), and glutamate ($r = 0.5412$, $P < 0.001$) (Fig. 1c, d). Glutamate levels were also significantly positively associated with HIV-1 DNA ($r = 0.3885$, $P = 0.002$, Fig. 1c, e). In further analysis of the potentially influential factors, glutamate levels were not influenced by ART-related factors, CD4 count, CD8 count, or CD4/CD8 ratio but by gender (Supplementary Table 2). Males exhibited significantly higher levels of plasma glutamate than females ($P = 0.020$; Supplementary Table 2). Moreover, the positive correlations between glutamate levels and the size of the HIV-1 reservoir were more apparent in males (HIV-1 CA usRNA, $r = 0.5884$, $P < 0.001$; HIV-1 DNA, $r = 0.4397$, $P = 0.006$) than in females (HIV-1 CA usRNA, $r = 0.4730$, $P = 0.031$; HIV-1 DNA, $r = 0.3560$, $P = 0.113$) (Supplementary Fig. 1a, b). These data suggest that higher plasma glutamate levels in PLWH indicate a more extensive HIV-1 reservoir.

**Glutamate levels were negatively correlated with the functionality of CD8 + T and TVM cells.** Recent studies have indicated that CD8+ T-cell function is significantly associated with viral control and delayed disease progression[4,28]. CD8+ T cells suppress HIV-1 replication through various effector molecules[7,29,30]. On this basis, we further investigated the effects of glutamate on the function of CD8+ T cells. In the correlation analyses, glutamate levels were negatively associated with multiple

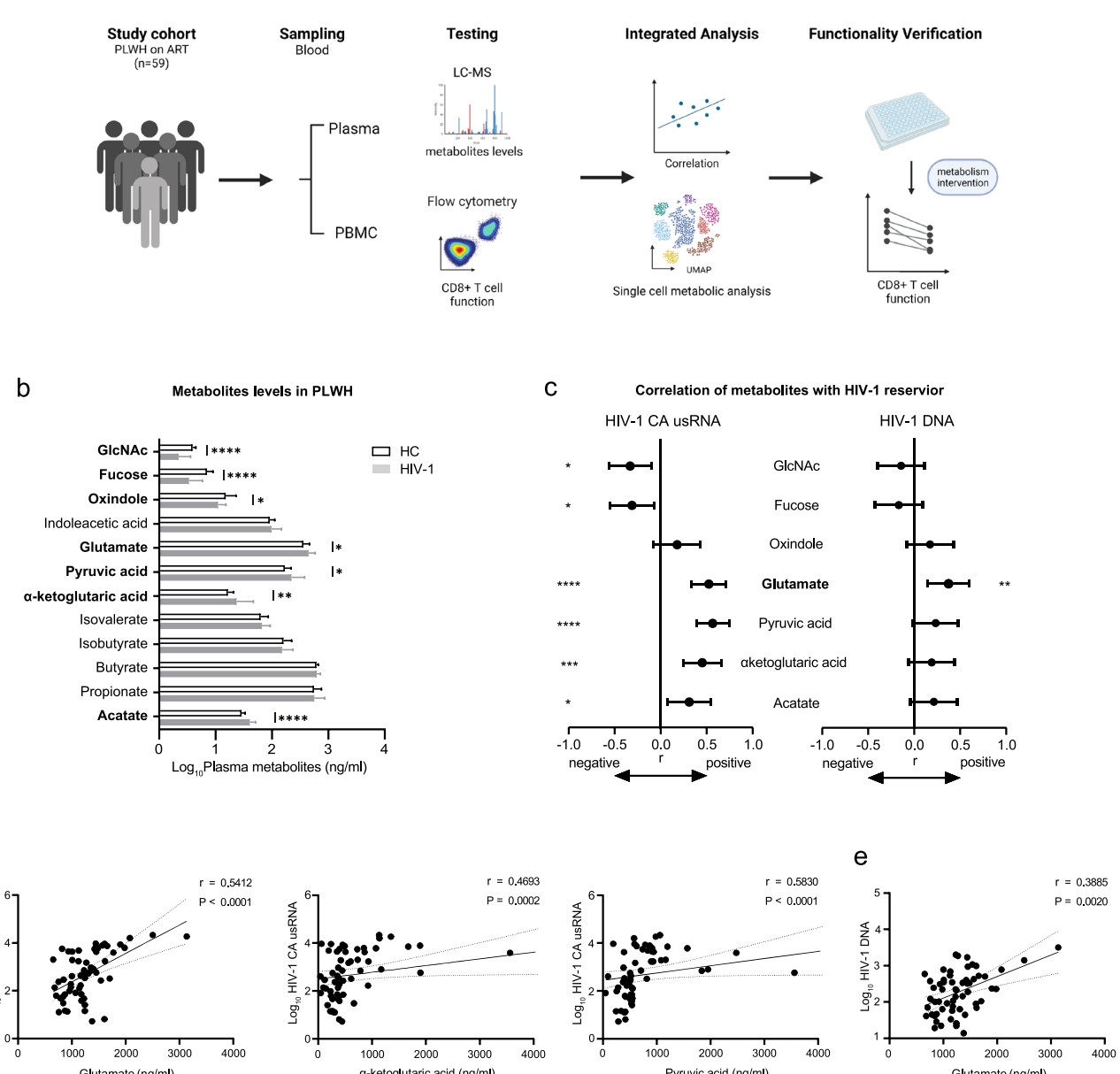

**Fig. 1 Glutamate positively correlated with HIV-1 reservoir size in ART individuals. a** Flowchart of study design (Created with BioRender.com.) **b** Comparison of alterations in plasma metabolites quantification in healthy control ($n = 11$) and PLWH ($n = 59$). The plot was shown as mean with standard deviation. **c** Correlations of metabolites levels with HIV-1 DNA and cell-associated CA usRNA levels. **d** Correlations of glutamate, α-ketoglutaric acid, and Pyruvic acid with and CA usRNA levels. **e** Correlations of glutamate with and HIV-1 DNA levels. The Mann–Whitney test (compare ranks) was used for the unpaired comparison. **b**, **c** correlations were evaluated using nonparametric Spearman correlation tests. **c** Black dots denote nonparametric Spearman $r$, and black lines denote 95% confidence interval. *$P < 0.05$, **$P < 0.01$, ***$P < 0.001$, and ****$P < 0.0001$.

cytokines and chemokines (Fig. 2a). Notably, glutamate levels negatively correlated with the proportion of CCL5 + CD8 + T ($r = -0.5808$, $P < 0.001$) and CCL3 + CD8 + T cells (Fig. 2b). Additionally, glutamate levels significantly negatively correlated with the proportions of CD107a + CD8 + T ($r = -0.4244$, $P = 0.003$) and IL-2 + CD8 + T cells ($r = -0.4925$, $P < 0.001$). Therefore, we further investigated the effects of glutamate on CD8 + T and TVM cell function.

Our previous work revealed that among CD8 + T cells, the TVM cell subset is superior in exerting anti-HIV function[23], depending on the secretion of CCL5[23,24]. As for TVM cells, glutamate levels were negatively associated with the proportions of CD107a+, IL-2+, CCL3+, and CCL5 + TVM cells (CD107a+ TVM cells,

$r = -0.3745$, $P = 0.010$; IL-2 + TVM cells, $r = -0.3993$, $P = 0.004$; CCL3 + TVM cells, $r = -0.3539$, $P = 0.023$; CCL5 + TVM cells, $r = -0.5311$, $P < 0.001$, Fig. 2c, d).

We noted that the CCL4+ fractions of CD8+ T and TVM cells showed the opposite tendency to CCL3 and CCL5 (Fig. 2a, c). Thus, we further analyzed the correlation between glutamate levels and the proportions of multifunctional CD8+ T and TVM cells. The proportion of CCL4 + CCL5- TVM cells positively correlated with glutamate levels, and that of CCL4- CCL5 + TVM cells negatively correlated with glutamate levels (CCL4 + CCL5- CCL3 + TVM cells, $r = 0.4899$, $P = 0.001$; CCL4 + CCL5- CCL3- TVM cells, $r = 0.4492$, $P = 0.003$; CCL4- CCL5 + CCL3 + TVM cells, $r = -0.3500$, $P = 0.025$; CCL4- CCL5 + CCL3- TVM

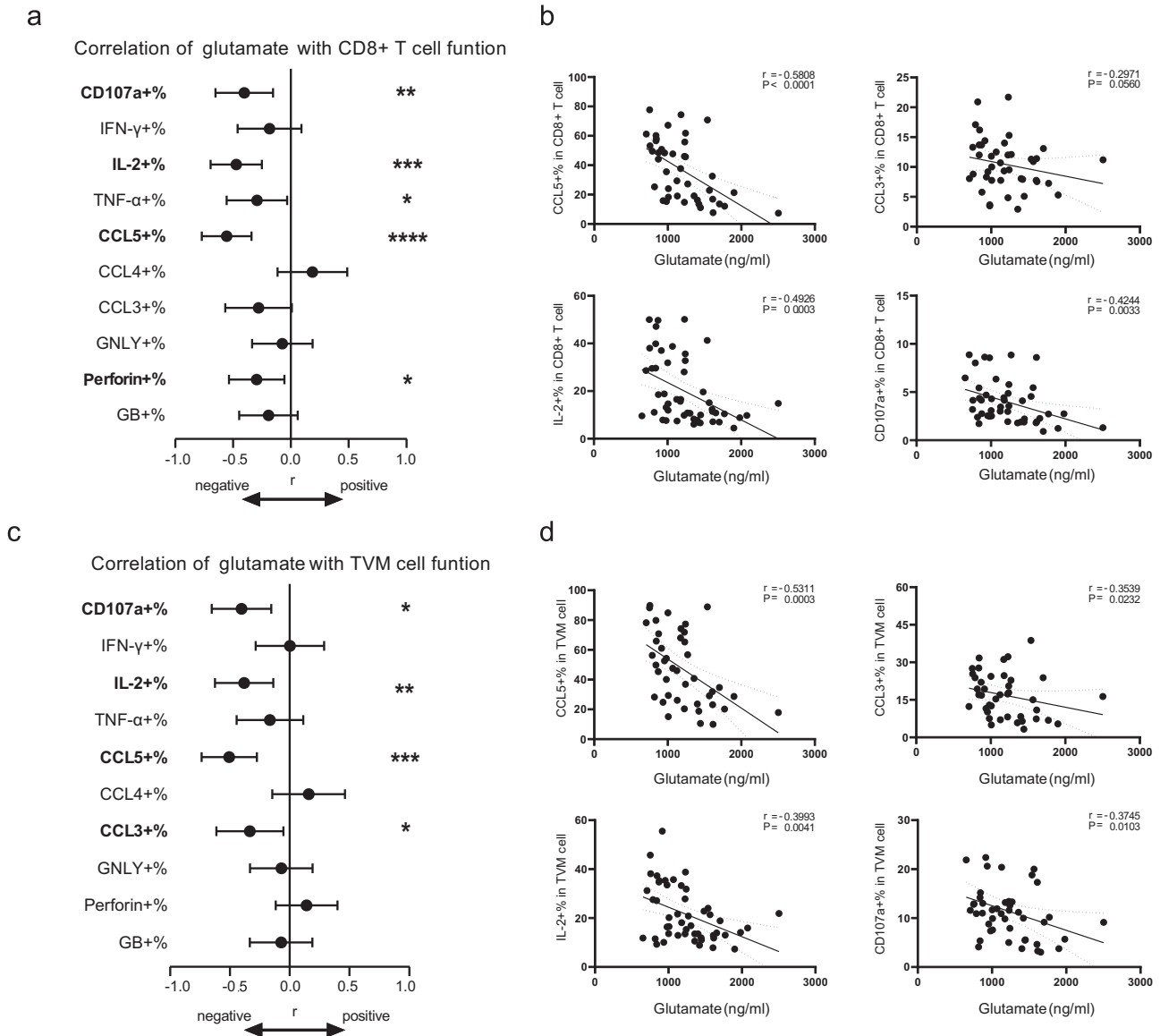

**Fig. 2 Glutamate negatively correlated with CD8+ T and TVM cell function. a** Correlations between percentages of cytokines producing CD8+ T cells and levels of plasma glutamate. **b** Correlations of CCL5+%, CCL3+%, IL-2+%, and CD107a+% in CD8+ T cell with plasma glutamate. **c** Correlations between percentages of cytokines producing TVM cells and plasma glutamate levels. **d** Correlations of CCL5+%, CCL3+%, IL-2+%, and CD107a+% in TVM cells with plasma glutamate. Correlations were evaluated using nonparametric Spearman correlation tests. *$P < 0.05$, **$P < 0.01$, ***$P < 0.001$ and ****$P < 0.0001$.

cells, $r = -0.4623$, $P = 0.002$). Similar correlative results were found between multifunctional CD8 + T cells and glutamate levels (Supplementary Fig. 2a, b). The heterogeneity among beta-chemokines might be explained by the fact that CCL4 + CD8 + T cells are reportedly associated with poor immune reconstitution in PLWH and may facilitate the establishment of the HIV-1 reservoir[31].

Together, these data suggest that glutamate might inhibit the function of CD8 + T cells and TVM cells.

**Single-cell transcriptional analysis of human TVM cells**. To further understand the transcriptional profile of CD8 + T cells among different disease states, we performed single-cell analysis using scRNA-seq dataset from an earlier study by our group[32]. Information on CD8 + T cells from five treatment-naïve (TN) PLWH, three ART-treated individuals, and four HCs was

extracted and subjected to downstream analysis (Supplementary Fig. 3a). After data quality control (Supplementary Fig. 3b), data normalization, merging, and dimensionality reduction analysis, 48654 CD8 + T cells were classified into nine clusters (Fig. 3a and Supplementary Fig. 3c), including two naïve subsets: naïve_1 (CCR7+ LEF1+) and naïve_ISG (IFI16+ IFI44L+); three CD8 + CM clusters: CM_1 (IL7R+ GPR183+), CM_2 (ITGB1+ S100A11+), and CM_CCL4 (CCL4 hi CTLA4+ HAVCR2+); two EMRA clusters: EMRA_1 (FGBP2+ GZMB+) and EMRA_KIR (TYROBP+ KIR2DL3+); EM_1 (GZMK+ RSG1+); and a MAIT cluster (SLC4A10 + KLRB1). The proportion of cells in the immune subpopulation of each individual is shown in Supplementary Fig. 3d.

EMRA_KIR was noted in TVM cells because of their high expression of KIR-related genes (KIR2DL3, KIR3DL1, KIR2DL1) and cytotoxic genes (GZMB, PFR1, GNLY) (Fig. 3a and Supplementary Fig. 3c). As for flow cytometric analysis, we used

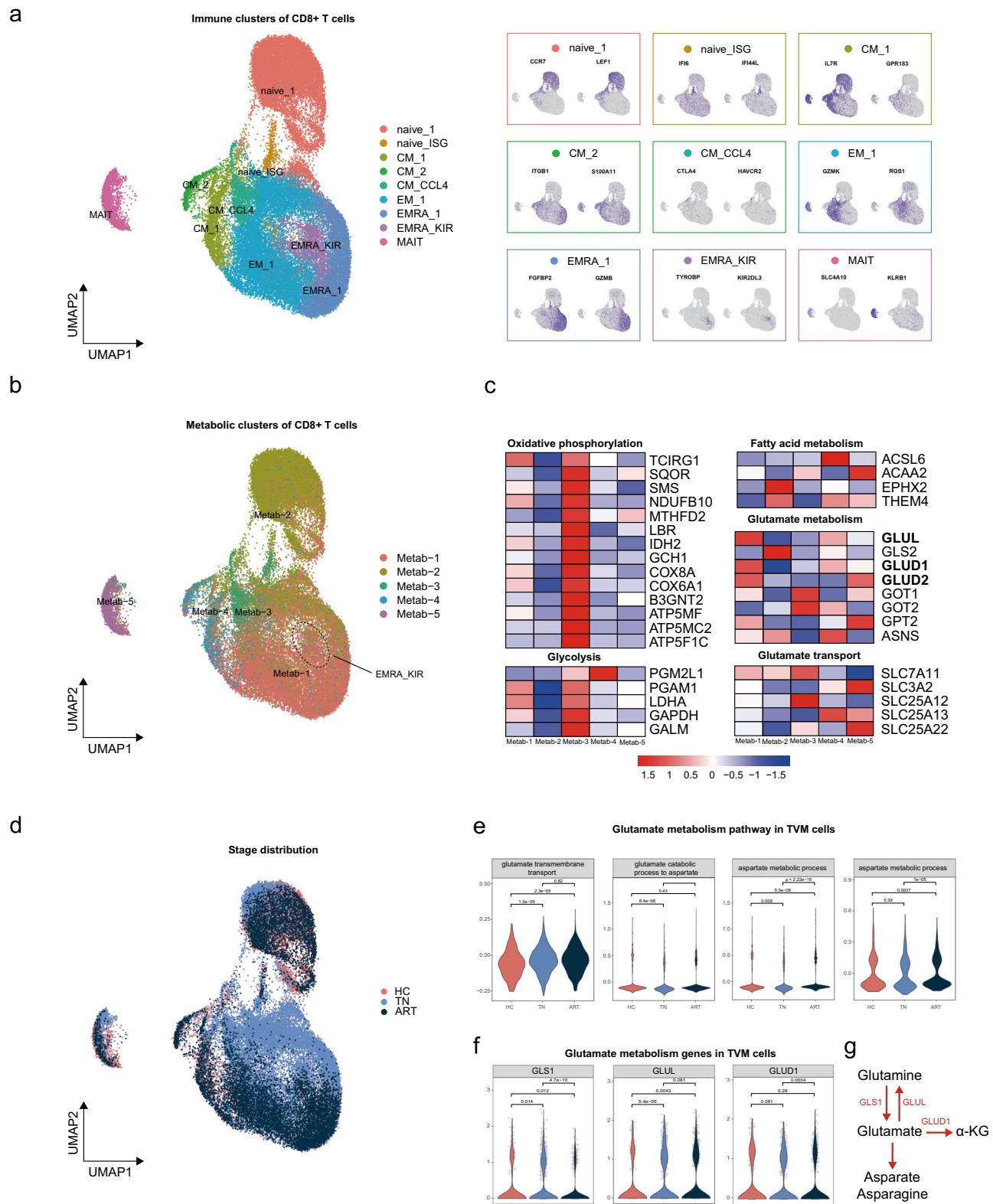

**Fig. 3 Glutamate metabolism in TVM cells was more active in the ART group than the TN group. a** Two-dimensional UMAP projection of 48654 cells was conducted by unsupervised clustering and the distribution of characteristic gene expression in each cluster. **b** UMAP projection of CD8+ T cell metabolic clusters. **c** Heatmap of glutamate metabolism genes and metabolism-related genes screened from DEGs. **d** Overlapping UMAP of CD8+ T cells in TN, ART, and HCs groups. **e** Violin plot comparing the score of glutamate-related metabolic pathways between TN, ART, and HC groups in TVM cells. **f** Violin plot comparing the expression of glutamate metabolism genes between TN, ART, and HC groups in TVM cells. **g** Summary of glutamate metabolism genes upregulated by the ART group.

pan-KIR/NKG2A positive in CD45RA+ CD8+ T cells as markers to identify TVM cells (Supplementary Fig. 4a)[33–35]. Since EOMES was another putative marker for human TVM cells, as reported by others[36,37], we also analyzed the expression of *EOMES* in CD8+ T cells. As expected, cells from EMRA_KIR cluster expressed high levels of *EOMES* (Supplementary Fig. 4b, c). Flow cytometric analysis confirmed that pan-KIR/NKG2A + CD45RA+ CD8+ T cells expressed higher levels of EOMES than total CD8+ T cells (Supplementary Fig. 4d). Moreover, CD45RA and EOMES were highly co-expressed on pan-KIR/NKG2A+ CD8+ T cells (Supplementary Fig. 4e).

In recent years, in-depth studies have revealed functional differences between KIR+ and NKG2A+ CD8+ T cells[38,39]. In our scRNA-seq dataset, expression of *KIR*-related genes was mainly restricted to the EMRA_KIR subset, while *KLRC1* (the gene encoded NKG2A) was expressed primarily on EMRA_KIR, CM_1, and MAIT subsets (Supplementary Fig. 5a, b). The expression of *KIRs* and *KLRC1* in EMRA_KIR cells was mutually exclusive (Supplementary Fig. 5c), akin to a recent study[39]. *KIRs*+ and *KLRC1*+ cells from the EMRA_KIR subset showed little differently expressed genes (DEGs; Supplementary Fig. 5d). However, *KLRC1*+ cells from EMRA_KIR significantly differed from *KLRC1*+ cells from CM_1, as evidenced by a large number of DEGs (Supplementary Fig. 5e). In particular, *KLRC1*+ cells from EMRA_KIR expressed higher levels of cytotoxic genes (*TYROBP, FGFBP2, NKG7, GZMB*, and *GZMH*), while *KLRC1*+ cells from CM_1 were more inflammatory (*LTB, GZMK, FOS*, and *JUN*) (Supplementary Fig. 5e). These data suggested that CD45RA+ NKG2A+ CD8+ T cells and CD45RA+ KIR+ CD8+ T cells, but not CD45RA- NKG2A+ CD8+ T cells, are functionally similar to TVM cells.

**CD8+ T cell subsets had distinct metabolic profiles**. To understand the metabolic characteristics of immune subpopulations, we performed unsupervised clustering analysis of CD8+ T cells based on the expression levels of metabolic genes reported by others[40–42]. CD8+ T cells were clustered into five subsets and overlapped well with the immune subpopulations (Fig. 3b, c). Specifically, Metab_1 mainly overlapped with EMRA_KIR, EMRA_1, and EM_1 clusters. Metab_1 expressed intermediate levels of oxidative phosphorylation (OXPHOS)- and glycolysis-related genes (NDUFB10 IDH2 COX8A) and high levels of glutamate metabolism genes (GLUL GLUD1 GLUD2), suggesting that Metab_1 is reserved for functional energy requirements and might be sensitive to variations in glutamate levels. Metab_2, mainly including Naïve-1, was characterized by low expression of OXPHOS and glycolysis-related genes and high expression of GLS (GLS2), indicating low levels of energy metabolism in this subset. CM_CCL4 and naïve_ISG were classified as Metab_3, which showed a high energy consumption state (high expression of OXPHOS and glycolysis-related genes). Metab_4, including CM_1 and CM_2, expressed fatty acid metabolism-related genes (ACSL6), thus exhibiting active fatty acid metabolism. Metab_5 consisted of MAIT cells and expressed high levels of SLC3A2, indicating high glutamate secretion activity (Fig. 3c). Among the five metabolic subsets, Metab_5 was decreased in the TN and ART groups compared to that in the HC group, whereas Metab_3 and Metab_1 were increased in TN and ART, respectively (Fig. 3d, Supplementary Fig. 6a). In summary, the metabolic profiles of immune subpopulations may be closely related to their functions. Our results suggest that TVM cells exhibit a unique metabolic profile, have active glutamate metabolism, and are capable of relying on both glycolysis and OXPHOS for energy supply.

**TVM cells from ART patients showed robust glutamate metabolism**. To study the metabolic plasticity of TVM cells across different groups and their connection with functionality, we defined several functional (cytotoxic, inflammatory, effector, and proliferation)[43] and metabolic (glutamate transmembrane transport, glutamate catabolism to aspartate, aspartate metabolic process, and asparagine metabolic process) scores (Supplementary Table 3). TVM cells in the ART group had higher cytotoxic and effector functions and lower inflammatory effects than those in the TN group (Supplementary Fig. 6b). Notably, TVM cells in the ART group exhibited a higher proliferative function than those in both the TN and HC groups (Supplementary Fig. 6b). Moreover, glutamate metabolism in TVM cells was the most active in the ART group (Fig. 3e), with high expression of *GLS1, GLUL*, and *GLUD1* (Fig. 3f, g). In conclusion, TVM cells in the ART group were hyperactivated, coinciding with robust glutamate metabolism.

**Glutamate inhibited TVM cell function ex vivo**. To validate the effect of glutamate on TVM cell function, we performed ex vivo experiments to confirm whether glutamate directly influences the antiviral function of TVM cells. We first assessed the impact of glutamate on cell antiviral function using peripheral blood mononuclear cell (PBMC) from PLWH receiving long-term ART. Cell-permeable form of glutamate (dimethyl DL-glutamate) was added to glutamine-free medium to study whether glutamate exerts an inhibitory effect on CD8+ T cells by affecting intracellular metabolic processes. The results showed that glutamate inhibited the overall production of IFN-γ, CCL5, and CD107a by CD8+ T and TVM cells from ART patients, respectively (Fig. 4a–d and Supplementary Fig. 7a, b). In addition, we confirmed that glutamate also inhibited the function of TVM cells from TN group (Supplementary Fig. 8a), with a higher efficiency than that of ART group (Supplementary Fig. 8b). This could be attributed to a greater activation of glutamate metabolism pathways in TVM cells from ART-treated PLWH compared to those from TN PLWH.

**Glutamate regulated TVM cell function through negative modulation of the mTORC1 pathway**. To further identify the mechanisms by which glutamate inhibits TVM cell function, we designed various interventions aimed at glutamate metabolic pathways as summarized in Fig. 5a. System Xc-, a Na+-dependent cystine-glutamate reverse transporter in the cytosol, is a disulfide-bonded heterodimer composed of SLC7A11 and SLC3A2 as the light and heavy chain subunits, respectively[44]. Erastin blocked the export of glutamate outside the cell by inhibiting system Xc- activity[45]. Interestingly, Erastin treatment, which blocked the extracellular transport of glutamate (Supplementary Fig. 9a), inhibited IFN-γ production by TVM cells, although CCL5 secretion was not significantly affected (Fig. 5b). Therefore, blocking the extracellular transport of glutamate might inhibit TVM cell function.

Accumulating evidence has highlighted the importance of glutamine metabolism in the regulation of T cell-mediated immunity[46–48]. Following certain intracellular transformations, glutamine can support cellular energy metabolism and regulate epigenetic reactions[49,50]. HIV-1 infection causes significant changes in glutamine metabolism in T cells, with an accompanying increase in glutamate secretion[51]. At the same time, glutaminase (GLS) deficiency increases the effector function of CD8 + CTL cells[52]. CB-839, a GLS inhibitor, was used to block the conversion of glutamine to glutamate. The data showed that IFN-γ and CCL5 production levels were not significantly affected

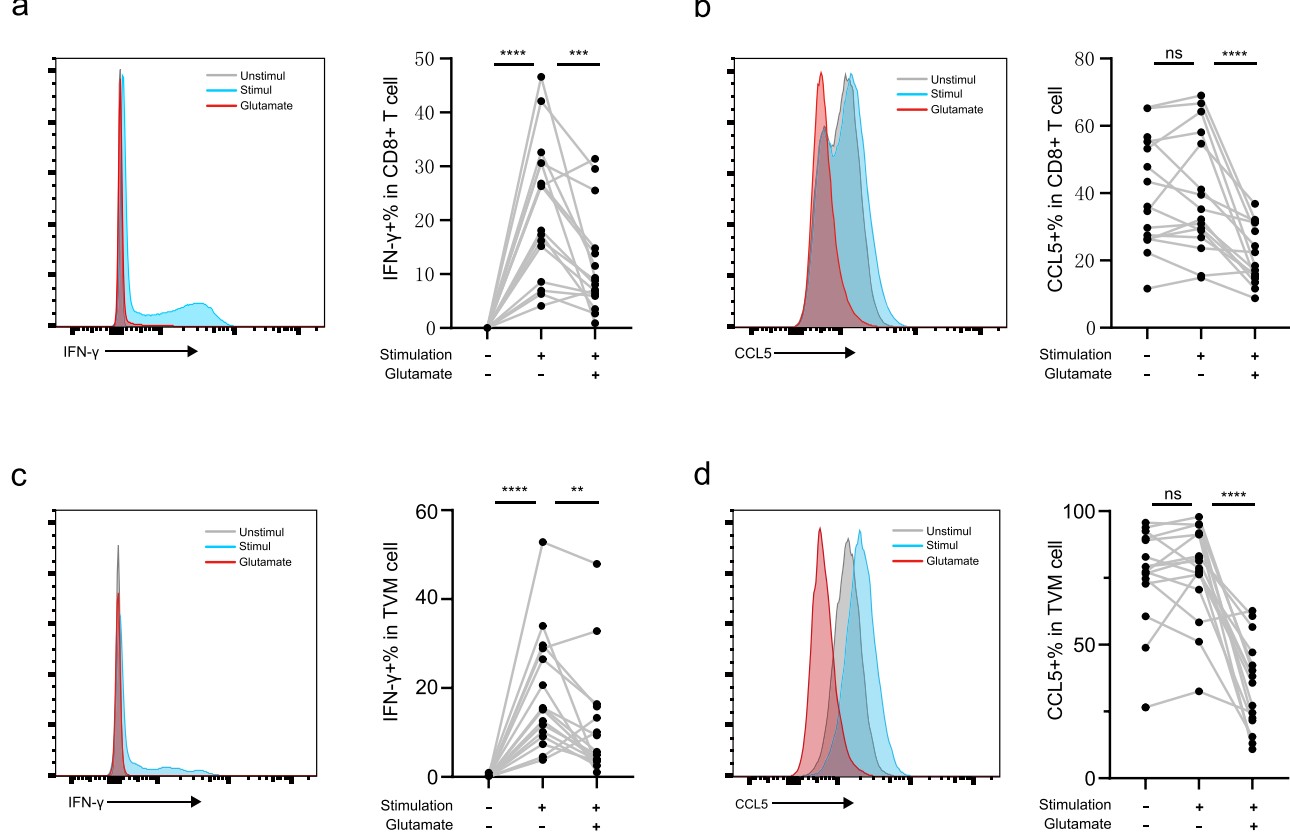

**Fig. 4 Glutamate inhibited CD8+ T and TVM cell function ex vivo. a** Representative IFN-γ expression in CD8+ T cell of unstimulated (gray), stimulated (blue) and 5 mm glutamate intervention(red). Summarized data show IFN-γ+% in CD8+ T cells following glutamate intervention. **b** Representative CCL5 expression in CD8+ T cell of unstimulated (gray), stimulated (blue), and 5 mm glutamate intervention (red). Summarized data show CCL5+% in CD8+ T cells following glutamate intervention. **c** Representative IFN-γ expression in TVM cell of unstimulated (gray), stimulated (blue), and 5 mm glutamate intervention(red). Variations in IFN-γ+% in TVM cell following glutamate intervention. **d** Representative CCL5 expression in TVM cell of unstimulated (gray), stimulated (blue), and 5 mm glutamate intervention (red). Summarized data show CCL5+% in TVM cells following glutamate intervention. DL-Glu, Dimethyl DL-Glutamate (hydrochloride). **a–d** $n = 16$ biologically independent experiments. The Mann–Whitney test (compare ranks) was used for the unpaired comparison; **$P < 0.01$, ***$P < 0.001$, and ****$P < 0.0001$ and ns not significantly.

in TVM cells (Supplementary Fig. 9b). In addition, we used L-methionine-DL-sulfoximine (MSO), an inhibitor of glutamine synthetase (GS), to inhibit the conversion of glutamate to glutamine. Similarly, this intervention had no significant effect on TVM cell function (Fig. 5c). Thus, the conversion of glutamate to glutamine by GLS and GS had a limited effect on the anti-HIV function of TVM cells.

Next, we examined the roles of glutamate-related metabolic pathways in glutamate-mediated inhibition of TVM-cell function. Glutamate can be converted to α-KG by glutamate dehydrogenase (GLUD), a process that antagonizes CD8 + T cell dysfunction through glutaminolysis and the tricarboxylic acid (TCA) cycle[53,54]. Treatment with GLUD inhibitors, including R162[55] or epigallocatechin gallate sulfate (EGCG)[56], did not rejuvenate but further inhibited the anti-HIV function of TVM cells (Fig. 5d and Supplementary Fig. 9c). These results suggest that the conversion of glutamate to α-KG promoted TVM-cell activation.

Glutamate also plays a central role in asparagine metabolism. Asparagine synthetase (ASNS) converts aspartate and glutamine to asparagine and glutamate, respectively[48]. Therefore, high levels of glutamate inhibit the synthesis of asparagine. Interestingly, Asn supplementation reversed the glutamate-mediated functional inhibition of TVM cells (Fig. 5e). Considering that both α-KG and Asn can act as metabolic activators via the mTORC1 signaling[53,57,58], we hypothesized that glutamate inhibited the TVM cell function by limiting mTORC1 activity

and downstream cellular metabolism. Glutamate treatment decreased the mTORC1 activity, as evidenced by decreased mTOR phosphorylation (Ser2448)[59] (Fig. 5f). Treatment with MHY-1485, a mTORC1 activator, effectively reversed the inhibitory effect of glutamate on TVM cell function (Fig. 5g and Supplementary Fig. 9d). In summary, glutamate suppressed the anti-HIV-1 function of TVM cells via the mTORC1 pathway in vitro (Fig. 5h).

## Discussion

HIV-1 infection induces profound metabolic alterations in the host, thereby reprograming the immune system. To elucidate the metabolites relevant to the functionality of CD8+ T cells and viral control in PLWH on ART, we quantified plasma metabolites using targeted LC-MS and analyzed their associations with the anti-HIV-1 activities of CD8+ T cells and HIV-1 reservoir size. We found that plasma glutamate levels were higher in PLWH on ART than in HCs, showing a positive correlation with viral reservoir size and a negative correlation with the CCL5-secreting ability of CD8+ T cells. Furthermore, glutamate inhibited the anti-HIV-1 function of CD8+ T cells via the mTORC1 pathway in vitro. These findings highlight glutamate-mediated repression of CD8+ T-cell function as a potential mechanism for HIV-1 persistence in PLWH.

Several metabolic-profiling cohort studies have shown that plasma glutamate levels are significantly higher in PLWH than in

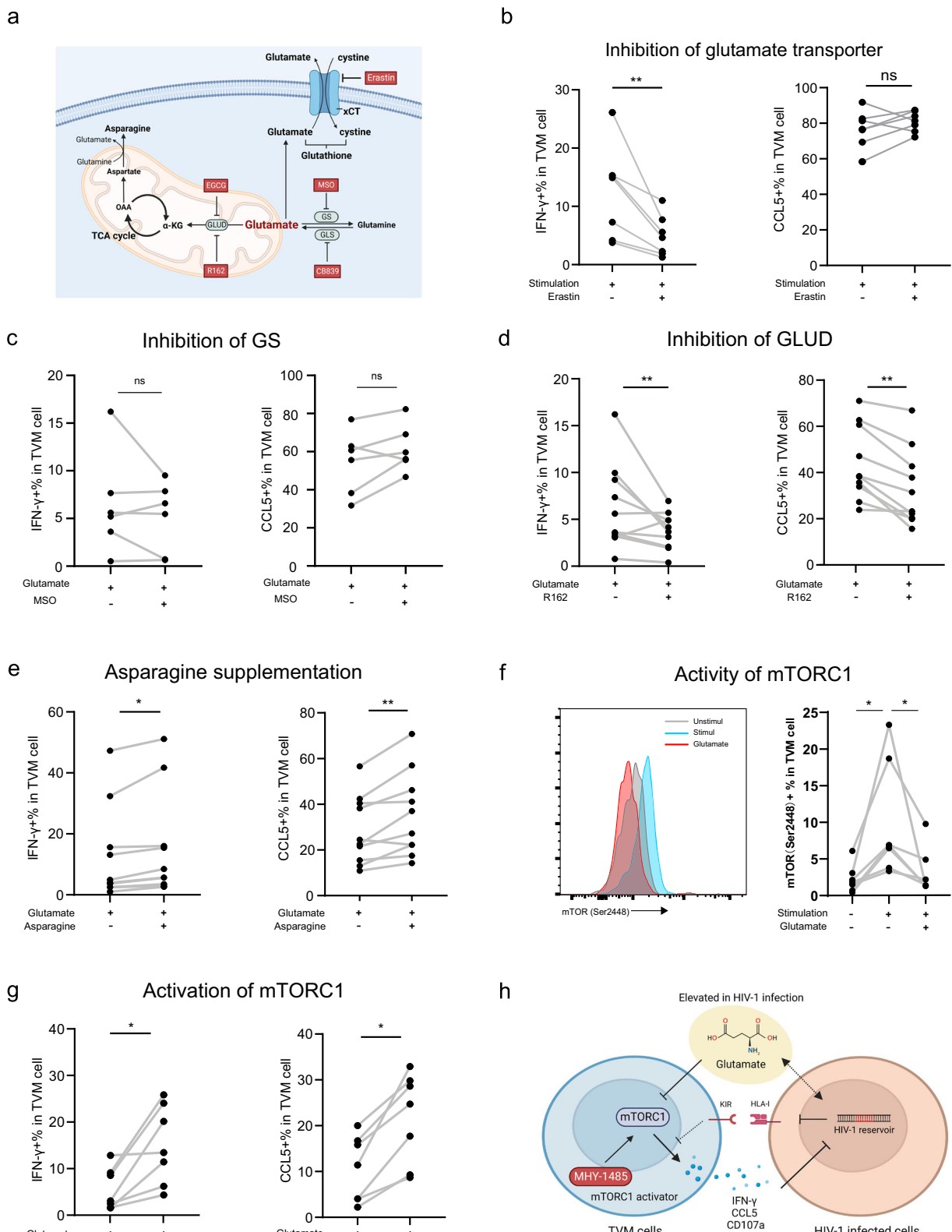

HCs[60–63]. Using a state-of-the-art targeted assay, we confirmed that plasma glutamate levels were considerably elevated in PLWH on ART compared to those in HCs. In a cohort from West Africa, PLWH on long-term antiretroviral therapy showed a trend toward the further elevation of plasma glutamate compared with untreated controls[20]. Moreover, glutamate levels were higher in the immune response (IR) group than in the immune

nonresponse (INR) group[64]. Similar to a previous report[65], we did not observe a significant association between plasma glutamate levels and CD4+ T cell counts. Altered glutamate levels are closely associated with HIV-1-related dementia and other neurological symptoms[66]. Furthermore, a recent study showed that alterations in glutamate metabolism are associated with metabolic syndrome (MetS) in PLWH[67]. Further studies on the clinical

**Fig. 5 Glutamate interfered with TVM cell function by inhibiting of the mTORC1 pathway. a** Glutamate metabolic pathways and intervention reagents (red box) involved in ex vivo experiments (Created with BioRender.com.). **b** Summarized data show IFN-γ+% and CCL5+% in TVM cells following Erastin intervention.. $n = 7$ biologically independent samples. **c** The impact of glutamate synthase inhibitor MSO on the proportion of IFN-γ+% and CCL5+% in TVM cells was assessed following glutamate intervention. $n = 6$ biologically independent samples. **d** The effects of glutamate intervention on the proportion of IFN-γ+% and CCL5+% in TVM cells were analyzed following treatment with glutamate dehydrogenase (GLUD) inhibitor R162. $n = 10$ biologically independent samples. **e** The effect of asparagine supplementation on the IFN-γ+% and CCL5+% in TVM cells after glutamate intervention. $n = 10$ biologically independent samples. **f** Impact of glutamate intervention on mTORC1 activity, with flow cytometry plot of mTOR(ser2448) expression and summary data. $n = 7$ biologically independent samples. **g** Changes in IFN-γ+% and CCL5+% in TVM cells after the treatment with mTORC1 activator MHY-1485 based on glutamate intervention. $n = 7$ biologically independent samples. **h** Summary diagram of how glutamate impacts the anti-HIV-1 function of TVM cells (Created with BioRender.com.). Wilcoxon matched-pairs signed rank tests (for two group comparisons) were performed. $*P < 0.05$, $**P < 0.01$ and ns not significantly. EGCG epigallocatechin gallate sulfate, MSO L-methionine-DL-sulfoximine.

relevance of elevated plasma glutamate levels in large cohorts are needed.

The plasma glutamate levels were significantly higher in males than in females. A positive correlation between glutamate levels and HIV-1 reservoir was observed in male patients but not in female patients. Interestingly, recent studies have reported that HIV-1 reservoirs exhibit sex-based differences; female patients had significantly fewer CD4+ T cells harboring HIV CA RNA and replication-competent virus, as measured by quantitative viral outgrowth assay[68,69]. The interplay between glutamate metabolism and HIV-1 persistence and its role in sex-based differences warrant further study.

Glutamate, an intermediate metabolite, and a crucial by-product of glutaminolysis, serves as an energy source along with glycolysis for the effector function of T cells[47–49]. By applying single-cell transcriptome analysis, we identified that TVM cells from ART-treated patients exhibited the highest glutamate metabolism activity among different CD8+ T cell subsets and disease stages. Further in vitro functional assays revealed that glutamate-mediated functional repression of TVM cells could not be explained by glutamate transporters or conversion to glutamine and αKG. However, Asn supplementation and stimulation with a mTORC1 agonist can compensate for glutamate-mediated functional repression in TVM cells. Further studies are required to understand the inhibitory effects of glutamate on the mTORC1 pathway.

Metabolites can also act directly on HIV-1 reservoir cells. HIV-1 reservoir tends to be established in metabolically active CD4+ T cells[17], which usually show a significant activation of glutaminolysis[26]. Studies have shown that mTORC1 regulates HIV-1 latency[70], and the use of mTORC1 inhibitors can inhibit the latent reversal of cells in patients with HIV-1[71]. Intervening with the HIV-1 reservoir by mediating the mTORC1 pathway during HIV-1 infection has the potential for clinical application[72]. Thus, targeting glutamate metabolism or mTORC1 provides multiple beneficial effects, such as restoring the anti-HIV-1 function of CD8+ T cells and enhancing HIV-1 production from reservoir cells.

TVM cells have innate adaptive immunity characteristics and exert quick effector functions, thus representing a crucial executioner for immune surveillance[73]. For instance, TVM cells have been implicated in various pathogenic conditions, including aging, infection, and cancer[23]. Research on human TVM cells is interesting but facing challenges, such as the difficulties in phenotypical characterization and heterogeneity of the cell subset[74]. Using single-cell transcriptional analysis and flow cytometric validation, we found that CD45RA and EOMES were highly co-expressed in panKIR and/or NKG2A positive CD8+ T cells, which unifies the two commonly used strategies to identify human TVM cells. Furthermore, we demonstrated the heterogeneity among NKG2A+ CD8+ T cells, which might explain the differences between KIR+ and NKG2A+ CD8+ T cells reported

in a recent study[39]. In particular, NKG2A+ TVM cells are phenotypically similar to KIR+ TVM cells, but are in sharp contrast to NKG2A+ non-TVM cells. Notably, NKG2A is also identified as an immune checkpoint for treating chronic infection and cancer[75,76]. These data provide insights into the phenotype and heterogeneity of human TVM cells.

Our study had several limitations. Owing to the cross-sectional design, we only measured the levels of metabolites in the patient's plasma at the time of sampling. Therefore, it was not possible to determine dynamic changes in metabolite levels and their association with disease status. In summary, our findings indicate that glutamate positively correlates with the HIV-1 reservoir and can inhibit CD8 + T cell function. It holds promise as a strategy for both intervening in the aberrant metabolic processes of PLWH and eliminating the latent HIV-1 reservoir to cure the disease.

## Method

**Study participants.** We enrolled patients with PLWH from the Fourth People's Hospital of Nanning, Nanning, China. The inclusion criteria for PLWH were as follows: (1) confirmed HIV-1 positive, successful ART > 2 years; (2) 18–65 years of age; (3) undetectable blood HIV-1 RNA < 20 copies/mL for at least 6–12 months; and (4) CD4 cell count >250/mL within seven days of clinical sample collection. The exclusion criteria were as follows: (1) occurrence of opportunistic infection; (2) co-infection with hepatitis B or C viruses; (3) pregnancy; (4) illicit intravenous drug use; (5) weight loss of > 10% in the past year; (5) body mass index (BMI) <18 or >25.0 kg/m$^2$; and (6) other common comorbidities, including seizure disorder and liver and kidney-related diseases. Additional individuals were screened using the same inclusion and exclusion criteria and enrolled for ex vivo functional assays. The study was approved by the the Ethics Committee of the Fifth Medical Center of Chinese PLA General Hospital 2016164D. All patients signed an informed consent form.

**Plasma-targeted metabolomics analysis.** LC-MS grade methanol and acetonitrile were purchased from Sigma (New Jersey, USA). Formic acid, 3-NPH (3-nitrophenyl trap), EDC, pyridine, and standard components were purchased from Aladdin Biochemical Technology Co. (Shanghai, China). According to the manufacturer's protocol, we placed 60 μm of plasma in a 2 mL centrifuge tube, added an equal volume of 50% acetonitrile-water, and vortexed the tube for 5 min. Subsequently, we mixed the sample with 60 μL of 200 mM 3-NPH in 75% methanol aqueous solution and 60 μL of 120 mM EDC (with 6% pyridine in methanol), placed the tube at rest at 40 °C for 1 h, and vibrated the tube once every 5 min. After completion of the reaction, the supernatant was centrifuged at 1600$g$ for 15 min at 4 °C and subjected to LC-MS analysis. Twenty milligrams of the standard was prepared in the same way and diluted with 50% ethacrynic acid to form a standard series of solutions with a concentration gradient. Chromatographic separation was performed using an Acquity UPLC system (Waters, Massachusetts, USA) with an HSS T3 column (2.1 mm × 100 mm, 1.8 μm, Waters). Chromatographic separation was performed at a column temperature of 40 °C and a flow rate of 0.30 mL/min. Mobile phase composition: A, water (0.01% formic acid) and B, acetonitrile (0.01% formic acid). Run time: 5 min; injection volume: 10 μL.

Mass data for plasma metabolites were obtained using a QTRAP 5500 QQQ mass spectrometer (SCIEX, Framingham, USA) equipped with an electrospray ion source in multiple reaction monitoring (MRM) modes. The parameters of MRM conversion, depolymerization potential (DP), and collision energy (CE) for all derived metabolites are shown in (Supplementary Table 4). Metabolite peak integration was performed using MultiQuant software (SCIEX). A standard curve was obtained by linear regression using the concentration of the standard as the

horizontal coordinate and the peak area as the vertical coordinate. The target metabolite concentration was calculated using a standard curve.

**Flow cytometry**. Peripheral blood was collected in ethylenediaminetetraacetic acid-anticoagulated tubes. Peripheral blood mononuclear cells (PBMCs) were isolated using Ficoll-Hypaque density-gradient centrifugation. Isolated PBMCs were stored in liquid nitrogen. PBMCs were resuspended and incubated in RPMI 1640 containing 10% fetal bovine serum (FBS) at 37 °C and 5% $CO_2$ for 2 h. CD8+ T and TVM cell functions were detected as described in our previous study[24]. Briefly, surface markers were stained with monoclonal antibodies (mAbs) for 30 min at 4 °C. For intracellular staining, samples were fixed and permeabilized using a Foxp3/transcription factor staining buffer set (Thermo Fisher Scientific Inc., Waltham, MA) and stained intracellularly with the indicated antibodies for 30 min at 4 °C. The stained samples were examined by flow cytometry on a BD Canto II flow cytometer (BD Biosciences, San Diego, CA, USA). Flow cytometry data were analyzed using FlowJo software (version 10.5.3; BD Biosciences). TVM cells were identified with the markers pan-KIR/NKG2A+ CD45RA+ in CD8+ T cells according to previous studies[23,24,33–35].

The following fluorescently conjugated antibodies or reagents were used for multicolor flow cytometry: APC/Fire 750 anti-CD3 (SK7), PerCP anti-CD8 (SK1), BV510 anti-CD45RA (HI100), PE-Cy7 anti-CD27 (O323), BV421 anti-perforin (dG9), APC anti-IL-2 (MQ1- 17H12), FITC anti-IFN-γ (4 S. B3), and BV421 anti-TNF-α (MAb11) from BioLegend (San Diego, CA); PerCP anti-CD4 (SK3), AF647 anti-GZMB (GB11), AF488 anti-GNLY (RB1), BV421 anti-CCL5 (2D5) and FITC anti-CD107a (H4A3) from BD Biosciences; PE anti-NKG2A (REA110), PE anti-KIR2D (DX27), PE anti-KIR3DL1 (DX9) and APC anti-NKG2A (REA110) from Miltenyi Biotec (Auburn, CA); FITC anti-CCL3 (CR3M) and AF647 anti-CCL4 (FL34Z3L) from Thermo Fisher Scientific Inc.; Live/dead dye was purchased from Thermo Fisher Scientific Inc.

**scRNA-seq analysis**. Primitive scRNA-seq data of CD8+ T cells were purified from PBMCs of three ART-treated PLWH, five treatment-naive PLWH, and four healthy individuals. The data were downloaded from the Genome Sequence Archive of the Beijing Institute of Genomics Data Center, Chinese Academy of Sciences [http://bigd.big.ac.cn/gsa-human, accession code HRA000190[32]]. The critical clinical features are shown in (Supplementary Fig. 3a). The gene expression matrix generating method and the parameters of the data quality control were the same as in a previous study[24](Supplementary Fig. 3b). The Scrublet package (v.0.2.3) in Python (v.3.9.7) was applied to each dataset to detect potential doublets with an expected doublet rate of 6%, and the cells with scores exceeding 0.5 were considered doublets and filtered out. The gene expression matrix was normalized using the NormalizeData function in the Seurat software.

The 12 scRNA-seq datasets were assembled into integrated datasets using the Seurat package. The elimination of batch effects and generation of uniform manifold approximation and projection (UMAP) were performed in the same manner as in our previous work[24]. In short, the integrated data set was generated with two functions (FindIntegrationAnchors and IntegrateData) and then subjected to dimensionality reduction and unsupervised clustering following the standard Seurat pipeline. In the principal component analysis, the total number of PCs was 50, the reduced dimensionality was set to 1–30 for clustering, and the resolution was 0.35.

The entire set of pathway genes was downloaded from the KEGG database (http://www.kegg.jp), and the gene sets of 85 pathways related to metabolism were selected. A total of 1716 metabolic genes in these pathways were used to substitute the previous high-variant genes and re-cluster all the cells. Finally, metabolic clusters were generated based on the original umap dimensions[40–42]. We used the FindMarkers function to calculate differentially expressed genes (DEGs) using the Wilcoxon rank-sum test and adjusted the P-values with Bonferroni correction. DEGs with a logarithmic fold change (log2FC) > 0.25 and adjusted P-values < 0.05 were selected and intersected with the metabolic gene set to obtain a total of 117 genes. These genes were grouped according to their metabolic pathways, the average expression was calculated, and heatmaps were drawn. Functional scores were calculated using the AddModuleScore function in the Seurat package, including both the KEGG and GO gene sets. The KEGG gene set was obtained from the forementioned KEGG website; the GO gene set was obtained from GSEA (https://www.gsea-msigdb.org/gsea/index.jsp). Rank-sum tests were performed to compare the differences in functional scores and gene expression between the groups.

**CD8+ T cell functional assay**. PBMCs were seeded at a density of $100 \times 10^4$ cells per well in 96-well plates. Different stimulation strategies were employed based on the experimental aim and target detection, as follows: For the production of GZMB, PRF, and GNLY, no stimulation was performed. For the production of CCL3, CCL4, and CCL5, cells were stimulated with 50 ng/mL IL-15 for 48 h, and 1 μg/mL anti-CD3, 1 μg/mL anti-CD28, 1 μg/mL anti-CD49d, and 500 IU/mL IL-2 were added 6 h before the end of stimulation. For the production of IL-2, IFNγ, TNFα and CD107a, cells were stimulated with 20 ng/mL IL-12, 10 ng/mL IL-15, and 20 ng/mL IL-18 for 48 h. Anti-CD107a antibody was added 6 h before the end of stimulation[77]. To block cytokine secretion, Golgi Plug was added 6 h before the end of stimulation. After stimulation, cells were collected and subjected to flow cytometry-based staining for cell surface antigen and intracellular antigen. Cell function was evaluated based on the expression of effector molecules.

**Ex vivo examination of the impact of glutamate metabolism on CD8+ T cell function**. We stimulated PBMCs with IL-15 (50 ng/mL) for 48 h (simultaneous addition of metabolites or metabolic intervention reagents), IL-2 (500 IU/mL), anti-CD3 (1 ng/mL), anti-CD28 (1 ng/mL), or anti-CD49d (1 ng/mL) for 6 h to detect intracellular CCL5 and IFN-γ expression. Golgi stop was simultaneously added to the medium 6 h before harvest to detect intracellular cytokines.

Recombinant human IL-2 and IL-15, anti-human CD3 (clone OKT3), anti-human CD28 (clone CD28.2), and anti-human CD49d (clone 9F10) antibodies were purchased from BioLegend. PRMI-1640 without L-glutamine was purchased from Sigma-Aldrich (R0883). The following metabolites and metabolic intervention reagents were used in the ex vivo experiments: dimethyl DL-glutamate (C3762, APEBIO, USA), L-glutamine (A8461, APEBIO, USA), Erastin (HY-15763, MCE, USA), CB-839 (B4799, APEBIO, USA), L-methionine-DL-sulfoximine (B8591, APEBIO, USA), R162 (HY-103096, MCE, USA), L-asparagine (HY-N0667, MCE, USA), and MHY1485 (SML0810, Sigma, USA). The concentrations of the metabolic intervention reagents used in the experiments were as follows: erastin, 10 μmol; CB-839, 1 μmol; L-methionine-DL-sulfoximine, 1 mmol; R162, 20 μmol; MHY-1485, 10 μmol.

**Statistics and reproducibility**. Based on the data distribution, log10 transformation was performed when the levels of multiple metabolites were compared. GraphPad Prism version 8.3.0 (GraphPad Software Inc., San Diego, CA, USA) was used for the statistical analysis. The histogram is shown as the mean with a 95% CI. For statistical comparison of the two groups, the Wilcoxon matched-pairs signed rank test was used for paired data, and the Mann–Whitney test (compare ranks) was used for unpaired data. Nonparametric Spearman's correlation analyses were used to examine the correlation between metabolites and other parameters. Statistical significance was set at $P < 0.05$.

**Reporting summary**. Further information on research design is available in the Nature Portfolio Reporting Summary linked to this article.

## Data availability

Main source data used to conduct scRNA-seq analysis can be accessed under the accession number http://bigd.big.ac.cn/gsa-human, accession code HRA000190. Source data are provided in Supplementary Data 1.

## Code availability

The source code used to reproduce our analysis can be accessed upon reasonable request from the corresponding author.

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

## Acknowledgements

This study was supported by the Outstanding Youth Training Fund of the Chinese PLA General Hospital (grant number 2019-JQPY-009), National Natural Science Foundation of China (grants 81721002), the Beijing Natural Science Foundation (No. 7222171), the Beijing Nova Program (No. 20220484135), and the National Innovation Group Project (No. 413FZT).

## Author contributions

F.-S.W. and C.Z. conceived the research, supervised the work performed, and worked with Y.-Y.W. and W.H. to construct the diagrams and write the manuscript. The participants were enrolled by Y.-J.L. and H.-H.H. The PBMC samples were collected and isolated by J.Z., S.-R.M., and Y.-Q.Q. Clinical data were collected by Y.-Y.W., W.H., and M.J.-Z. Flow cytometry experiments were performed by Y.-Y.W., W.H., J.L., M.-J.Z., Y.-L.F., X.-Y.L., and T.Y. with technical support of C.-B.Z. and J.-H.Y. ScRNA-seq data was analyzed by C.Z. and P.Z.; J.-W.S., X.F., Y.-M.J., R.-N.X., J.-Y.Z., L.C., and L.H. edited the manuscript and provided comments and feedback. F.-S.W., and M.S. managed the project team and supervised data collation. All authors read and approved the final manuscript.

## Competing interests

The authors declare no competing interests.
