## [Peer Review File · Communications Biology]

Reviewers' comments:

Reviewer #1 (Remarks to the Author):

In this manuscript, the authors first confirmed that glutamate can inhibit the anti-HIV-1 function of TVM cells via the mechanistic target of rapamycin complex 1 (mTORC1) pathway in vitro, which means that glutamate metabolism can be exploited as a therapeutic target for the reversion of anti-HIV CD8+ T cell function in PLWH.

Major point

1 In result 4, authors had confirmed that compared to TN and HC groups, TVM cells in the ART group were hyperactivated, coinciding with robust glutamate metabolism. Is this related to the treatment? I think in result 5, you should have another experiment which can validate the effect of glutamate on TVM cell function in PBMC from treatment-naïve PLWH. Then we can know that if glutamate can really inhibit TVM cell function in vivo, or only can inhibit TVM cell function in ART group.

2 In result 5, authors used Erastin to block the export of glutamate outside the cell by inhibiting system Xc activity, but they didn't show the concentration of glutamate in the cytoplasm. Similarly, CB-839, an inhibitor of glutamine synthetase, MSO, an inhibitor of glutamine synthetase, R162, a selective inhibitor of GLUD, these experiments should also have the concentrations of glutamate in the cytoplasm.

3 In result 5, authors hypothesized that glutamate inhibited the TVM cell function by limiting mTORC1 activity and downstream cellular metabolism, however there aren't experiments can show this.

4 In result 5, authors used MHY-1485 to activate mTORC1, which can reversed the inhibitory effect of glutamate on TVM cell function. But there aren't experiment can show the activity of mTORC1.

Minor point

1 The symbols In Figure1B is not clear.

2 In line137, 'CD8+ T cells control HIV-1' is not accurate.

3 Supplement Figure 2A font sizes seem to be inconsistent.

4 In line 213 and line 215, Supplement Figure 3E should be Supplement Figure 3F.

Reviewer #2 (Remarks to the Author):

Several publication on the role of human TVM cells started to arise in the last 3 years bringing new information on their biology and role in different pathological contexts. In this matter, the work by Wang YY et al. contribute to the knowledge on human EMRA+ KIRs+ CD8+ cells in the context of a HIV viral infection. Moreover, they report how the metabolic state (especially glutamate) could condition the functional status of CD8+ cells. Additionally, the data on glutamate inhibiting the EMRA+ KIRs+ CD8+ cells function by limiting mTOR activity is a relevant information in the field. The work is solid however; some crucial concerns discourage their results especially pointing out TVM cells as the major players in HIV infection.

Major concerns:

CD45RA and CD27 are not exclusive as a panel to discriminate human TVM cells as it has been reported that both markers could be expressed or not by TEMRA+ of TVM cells (Warren et al, J. Leuk Biol. 79:1252. 2006, Jacomet et al, DOI: 10.1002/eji.201545539, Daniel et al. 2021, doi: 10.3389/fimmu.2021.674016).

Based on reference 24 (previous publication of the authors) and in the present manuscript, Eomes is not used as part of the consensus lineage markers of human TVM (Jacomet et al, DOI:

10.1002/eji.201545539).

Furthermore: The authors state in M&M: TVM cells were selected based on CD45RA+CD27 – TEMRA cells according to whether they expressed KIR or NKG2A. However, in the paper they seem to analyze KIRs+ cells (Figure 3A and suppl. fig. 3A). This is an important point to take into consideration since lately, KIRs+ TVM and NKG2DA+ TVM cells have been distinguished in 2 populations that carry very different functional characteristics (Pieren et al, 2021, DOI: 10.1111/ace.13372)

It is indispensable to incorporate Eomes marker in order to call these cells as TVM or to change the name in the manuscript as KIRs/NKG2D+ EMRA+ CD8+ T cells.

Minor concerns:

Taking into account that glutamate is able to suppress TVM? cells function (only measured by IFN γ and CCL5 expression). It would have been interesting also to include CD107a evaluation in the functional panel of figures 4 and 5.

Moreover, CD107a assays determined in figure 2 are not described in the M&M section. Do the authors use PMA/Ionomycin stimulation for detection of this marker, or anti-CD16 plate-coated stimulation or co-culture with K562 target cells as previously reported for TVM cells (Jacomet et al, DOI: 10.1002/eji.201545539)

Reviewers' comments:

Reviewer #1 (Remarks to the Author):

In this manuscript, the authors first confirmed that glutamate can inhibit the anti-HIV-1 function of TVM cells via the mechanistic target of rapamycin complex 1 (mTORC1) pathway in vitro, which means that glutamate metabolism can be exploited as a therapeutic target for the reversion of anti-HIV CD8+ T cell function in PLWH.

Major point

Q1 In result 4, authors had confirmed that compared to TN and HC groups, TVM cells in the ART group were hyperactivated, coinciding with robust glutamate metabolism. Is this related to the treatment? I think in result 5, you should have another experiment which can validate the effect of glutamate on TVM cell function in PBMC from treatment-naïve PLWH. Then we can know that if glutamate can really inhibit TVM cell function in vivo, or only can inhibit TVM cell function in ART group.

Response: Thank the reviewer for the instructive suggestions. Per your suggestion, we have performed experiments to test the effect of glutamate on TVM cell function in PBMCs from treatment-naïve PLWH (n = 12). Data showed that glutamate treatment inhibited the production of IFN- γ and CCL5 in TVM cells from TN patients (**Figure S8a**). Moreover, the inhibition efficiency was slightly higher in TNs than that in ART patients (**Figure S8b**, IFN- γ , TN: 78.79 \pm 11.92, ART: 59.49 \pm 23.24, p = 0.0168; CCL5, TN: 77.99 \pm 18.45, ART: 56.66 \pm 19.94, p = 0.0068), which might be due to that the activities of glutamate metabolism pathways were more active in TVM cells from ART patients than that in TN patients. However, whether these differences were related to the treatment was not clarified by these data, because complexity of factors influencing TVM cell function, such as aging, persistent inflammation and medication, which warrants further study. We have supplemented these data and discussions in the revised manuscript.

Q2 In result 5, authors used Erastin to block the export of glutamate outside the cell by inhibiting system Xc activity, but they didn't show the concentration of glutamate in the cytoplasm. Similarly, CB-839, an inhibitor of glutamine synthetase, MSO, an inhibitor of glutamine synthetase, R162, a selective inhibitor of GLUD, these experiments should also have the concentrations of glutamate in the cytoplasm.

Response: Thank you for pointing this out. We are sorry we were not able to conduct intracellular monitoring of glutamate metabolism for these interventions due to lack of the related devices in our laboratory. Alternatively, we have performed additional experiments for the quality control of these metabolic interventions. (1) We detected the concentrations of glutamate in the cell culture

supernatant with or without Erastin treatment^{1, 2}. Data showed that Erastin treatment resulted in a decrease of glutamate levels in the supernatant (**Figure S9a**), which could be due to Erastin-induced blocking of glutamate efflux; (2) We used another inhibitor, epigallocatechin gallate sulfate (EGCG), for GLUD inhibition. EGCG treatment inhibited TVM cell function (**Figure S9c**), which was consistent with R162 treatment³. In addition, these metabolic interventions have been widely used for similar purposes in earlier studies^{4, 5, 6}, we have supplemented the related information in the revised manuscript.

1. Liu N, Lin X, Huang C. Activation of the reverse transsulfuration pathway through NRF2/CBS confers erastin-induced ferroptosis resistance. *British journal of cancer* 122, 279-292 (2020).
2. Zille M, et al. Ferroptosis in neurons and cancer cells is similar but differentially regulated by histone deacetylase inhibitors. *Eneuro* 6, (2019).
3. Choi Y-K, Park K-G. Targeting glutamine metabolism for cancer treatment. *Biomolecules & therapeutics* 26, 19 (2018).
4. Varghese S, et al. The Glutaminase Inhibitor CB-839 (Telaglenastat) Enhances the Antimelanoma Activity of T-Cell-Mediated Immunotherapies. *Mol Cancer Ther* 20, 500-511 (2021).
5. Best SA, et al. Glutaminase inhibition impairs CD8 T cell activation in STK11-/Lkb1-deficient lung cancer. *Cell Metab*, (2022).
6. Albrecht J, Norenberg MD. L-methionine-DL-sulfoximine induces massive efflux of glutamine from cortical astrocytes in primary culture. *European journal of pharmacology* 182, 587-590 (1990).

Q3 In result 5, authors hypothesized that glutamate inhibited the TVM cell function by limiting mTORC1 activity and downstream cellular metabolism, however there aren't experiments can show this.

Response: Thank you for pointing this out. We have measured Serine 2448 (Ser2448) phosphorylation as a proxy of mTOR kinase activity¹. Data showed that glutamate treatment decreased the levels of Ser2448 phosphorylation (Figure 5f).

1. Argyriou P, *et al.* Hypoxia-inducible factors in mantle cell lymphoma: implication for an activated mTORC1 → HIF-1 α pathway. *Annals of hematology* 90, 315-322 (2011).

Q4 In result 5, authors used MHY-1485 to activate mTORC1, which can reversed the inhibitory effect of glutamate on TVM cell function. But there aren't experiment can show the activity of mTORC1.

Response: Thank you for pointing this out. Similar to the response to Q3, we found that the levels of mTOR phosphorylation (Ser2448) increased after MHY1485 treatment (Figure S9d).

Minor point

1 The symbols In Figure1B is not clear.

Response: We have modified the plotting of Figure 1B in the revision.

2 In line137, 'CD8+ T cells control HIV-1' is not accurate.

Response: We have changed "CD8+ T cells control HIV-1" to "CD8+ T cells suppress HIV-1 replication" in the revision.

3 Supplement Figure 2A font sizes seem to be inconsistent.

Response: We have modified in the revision.

4 In line 213 and line 215, Supplement Figure 3E should be Supplement Figure 3F.

Response: Thanks for pointing out. We have modified in the revision.

Reviewer #2 (Remarks to the Author):

Several publications on the role of human TVM cells started to arise in the last 3 years bringing new information on their biology and role in different pathological contexts. In this matter, the work by Wang YY et al. contribute to the knowledge on human EMRA+ KIRs+ CD8+ cells in the context of a HIV viral infection. Moreover, they report how the metabolic state (especially glutamate) could condition the functional status of CD8+ cells. Additionally, the data on glutamate inhibiting the EMRA+ KIRs+ CD8+ cells function by limiting mTOR activity is a relevant information in the field.

The work is solid however; some crucial concerns discourage their results especially pointing out TVM cells as the major players in HIV infection.

Major concerns:

CD45RA and CD27 are not exclusive as a panel to discriminate human TVM cells as it has been reported that both markers could be expressed or not by TEMRA+ or TVM cells (Warren et al, J. Leuk Biol. 79:1252. 2006, Jacomet et al, DOI: 10.1002/eji.201545539, Daniel et al. 2021, doi: 10.3389/fimmu.2021.674016). Based on reference 24 (previous publication of the authors) and in the present manuscript, Eomes is not used as part of the consensus lineage markers of human TVM (Jacomet et al, DOI: 10.1002/eji.201545539). Furthermore: The authors state in M&M: TVM cells were selected based on CD45RA+CD27 – TEMRA cells according to whether they expressed KIR or NKG2A. However, in the paper they seem to analyze KIRs+ cells (Figure 3A and suppl. fig. 3A). This is an important point to take into consideration since lately, KIRs+ TVM and NKG2DA+ TVM cells have been distinguished in 2 populations that carry very different functional characteristics (Pieren et al, 2021, DOI: 10.1111/ace.13372). It is indispensable to incorporate Eomes marker in order to call these cells as TVM or to change the name in the manuscript as KIRs/NKG2D+ EMRA+ CD8+ T cells.

Response: Thank the reviewer for pointing out this critical issue. We agree with the reviewer that using the term TVM cell might lead to ambiguity in this study. First, the TVM cell is characterized by memory features without encountering cognate antigen, which was well-defined in mice but hard to verify in humans; Second, the putative human TVM population seems to constitute heterogeneous subgroups and shows overlap with many other CD8+ T cell subsets, such as KIR+ regulatory CD8+ T cell, innate memory CD8+ T cell and NK-like CD8+ T cell. However, we would like to persevere in the use of TVM cell in the revision after carefully addressing the reviewer's concerns.

(1) About the flow cytometric gating strategy for human TVM cells

With much efforts from pioneers in the field, several methods have been developed to identify human TVM cells, such as KIR and/or NKG2A positive in CD45RA+ CD8+ T cells^{1, 2, 3}, as well as KIR and/or NKG2A positive in EOMES+ CD8+ T cells^{4, 5, 6}. In accordance with our previous studies^{7, 8}, we used panKIR/NKG2A+ CD45RA+ CD8+ as the markers to identify human TVM

cells. We are sorry that CD27 was mislabeled in the gating strategy in the previous Fig S4A, which was corrected in the revision.

As mentioned by the reviewer, EOMES⁺ was another putative marker for human TVM cells, which was not involved in our analysis. Accordingly, our data confirmed that the pan-KIR/NKG2A⁺ CD45RA⁺ CD8⁺ T cells expressed higher levels of EOMES⁺ than total CD8⁺ T cells at both transcriptional (**Figure S4b, c**) and proteomic (**Figure S4d**) levels. Moreover, CD45RA and EOMES were highly co-expressed in pan-KIR/NKG2A⁺ CD8⁺ T cells (**Figure S4e**). Thus, our data support the use of either CD45RA or EOMES in addition to panKIR/NKG2A to identify human TVM cells.

(2) About the heterogeneity of KIR⁺ and NKG2A⁺ CD8⁺ T cells

As shown in Figure 3, unbiased clustering of CD8⁺ T cells based on scRNA-seq data revealed nine different subclusters. Among these, the expression of KIR genes was mainly restricted to the EMRA_KIR subset, while *KLRC1* (the gene encoded NKG2A) was expressed primarily on EMRA_KIR, CM_1, and MAIT subsets (**Figure S5a, b**). Notably, the expression of *KIRs* and *KLRC1* in EMRA_KIR cells was mutually exclusive (**Figure S5c**), akin to a recent study⁹. *KIR*⁺ and *KLRC1*⁺ cells from the EMRA_KIR subcluster showed little differently expressed genes (DEGs) (**Figure S5d**). However, *KLRC1*⁺ cells from EMRA_KIR remarkably differed from *KLRC1*⁺ cells from CM_1, as evidenced by a large number of DEGs (**Figure S5e**). *KLRC1*⁺ cells from EMRA_KIR expressed higher levels of cytotoxic genes (*TYROBP*, *FGFBP2*, *NKG7*, *GZMB*, and *GZMH*), while *KLRC1*⁺ cells from CM_1 were more inflammatory (*LTB*, *GZMK*, *FOS*, and *JUN*) (**Figure S5e**). These data indicated that CD45RA⁺ NKG2A⁺ CD8⁺ T cells and CD45RA⁺ KIR⁺ CD8⁺ T cells, but not CD45RA⁻ NKG2A⁺ CD8⁺ T cells, are functionally similar to TVM cells.

1. White JT, et al. Virtual memory T cells develop and mediate bystander protective immunity in an IL-15-dependent manner. *Nat Commun* 7, 11291 (2016).
2. Quinn KM, et al. Age-Related Decline in Primary CD8(+) T Cell Responses Is Associated with the Development of Senescence in Virtual Memory CD8(+) T Cells. *Cell Rep* 23, 3512-3524 (2018).
3. Quinn KM, et al. Metabolic characteristics of CD8(+) T cell subsets in young and aged individuals are not predictive of functionality. *Nat Commun* 11, 2857 (2020).
4. Jacomet F, et al. Evidence for eomesodermin-expressing innate-like CD8(+) KIR/NKG2A(+) T cells in human adults and cord blood samples. *Eur J Immunol* 45, 1926-1933 (2015).
5. Barbarin A, et al. Phenotype of NK-like CD8 (+) T cells with innate features in humans and their relevance in cancer diseases. *Frontiers in immunology* 8, 316 (2017).
6. Kasakovski D, et al. Characterization of KIR + NKG2A + Eomes- NK-like CD8+ T cells and their decline with age in healthy individuals. *Cytometry B Clin Cytom* 100, 467-475 (2021).
7. Jin JH, et al. Virtual memory CD8+ T cells restrain the viral reservoir in HIV-1-infected patients with antiretroviral therapy through derepressing KIR-mediated inhibition. *Cell Mol Immunol* 17, 1257-1265 (2020).

8. Hu W, et al. CCL5-Secreting Virtual Memory CD8+ T Cells Inversely Associate With Viral Reservoir Size in HIV-1-Infected Individuals on Antiretroviral Therapy. *Front Immunol* 13, 897569 (2022).
9. Choi SJ, et al. KIR(+)CD8(+) and NKG2A(+)CD8(+) T cells are distinct innate-like populations in humans. *Cell Rep* 42, 112236 (2023).

Minor concerns:

Taking into account that glutamate is able to suppress TVM? cells function(only measured by IFN γ and CCL5 expression). It would have been interesting also to include CD107a evaluation in the functional panel of figures 4 and 5.

Response: Thank the reviewer for the suggestion. We have performed additional experiments and confirmed that glutamate treatment inhibited the expression of CD107a in both CD8+ T cells and TVM cells (**Figure S7a, b**). Moreover, the inhibitory effects of glutamate of CD107a expression were equal in KIR+ and NKG2A+ TVM cells (**Response Fig 1**).

Response Figure 1. Effect of glutamate intervention on the expression of CD107a in KIR+ and NKG2A+ TVM cells. (A) Alterations of CD107a expression in KIR+ TVM cells after glutamate intervention. (B) Alterations of CD107a expression in NKG2A+ TVM cells after glutamate intervention. (C) Comparison of the inhibition rate of CD107a expression in KIR+ and NKG2A+ CD8+ T cells. *P < 0.05.

Moreover, CD107a assays determined in figure 2 are not described in the M&M section. Do the authors use PMA/Ionomycin stimulation for detection of this marker, or anti-CD16 plate-coated stimulation or co-culture with K562 target cells as previously reported for TVM cells (Jacomet et al, DOI: 10.1002/eji.201545539)

Response: We are sorry for the confusion caused by the unclarity. In the CD107a assays, cells were stimulated with 20 ng/mL IL-12, 10 ng/mL IL-15, and 20 ng/mL IL-18 for 48 h, and anti-CD107a antibody was added 6 h before the end of stimulation^{1,2}. We have added this information in the revision.

1. Hu W, et al. CCL5-Secreting Virtual Memory CD8⁺ T Cells Inversely Associate With Viral Reservoir Size in HIV-1-Infected Individuals on Antiretroviral Therapy. *Front Immunol* 13, 897569 (2022).
2. Boieri M, et al. IL-12, IL-15, and IL-18 pre-activated NK cells target resistant T cell acute lymphoblastic leukemia and delay leukemia development in vivo. *Oncoimmunology* 6, e1274478 (2017).

REVIEWERS' COMMENTS:

Reviewer #1 (Remarks to the Author):

The revised manuscript has addressed the concerns raised by the reviewers, and is recommended for publication.

Reviewer #2 (Remarks to the Author):

The authors have correctly responded to the reviewer's request and added new additional data that support their hypothesis.